# Hidden Markov model for acoustic pesticide exposure detection and hive identification in stingless bees

Alex Otesbelgue[1]*, Amara Jean Orth[2], Chandler David Fong[3], Carol Anne Fassbinder-Orth[4], Betina Blochtein[5], Maria João Ramos Pereira[1]

1 Graduate Program in Ecology, Institute of Biosciences, Universidade Federal do Rio Grande do Sul, Porto Alegre, Rio Grande do Sul, Brazil, 2 Department of Earth and Planetary Sciences, Stanford University, Stanford, California, United States of America, 3 Baylor College of Medicine, Houston, Texas, United States of America, 4 Biology Department, Creighton University, Omaha, Nebraska, United States of America, 5 Mais Abelhas Consultoria Ambiental Co., Porto Alegre, Rio Grande do Sul, Brazil

* alex.otesbelgue@gmail.com

## Abstract

Pollinator populations are declining globally at an unprecedented rate, driven by factors such as pathogens, habitat loss, climate change, and the widespread application of pesticides. Although colony losses remain difficult to prevent, precision beekeeping has introduced non-invasive strategies for monitoring hive conditions. Acoustic data, combined with machine learning techniques, has proven effective in detecting stressors and specific events in honeybee colonies; however, such methodologies remain underexplored for stingless bees, a group of native pantropical pollinators. Meliponiculture, the practice of keeping stingless bees, is an expanding field that offers significant economic and conservation benefits. Stingless bees are particularly susceptible to pesticide toxicity, even at residual concentrations, underscoring the critical need to prevent hive losses and to understand the impacts of sub-lethal pesticide exposure on these species. This study addresses the challenge of detecting airborne pesticide exposure by aiming to identify stress responses in hives of the stingless bee *Tetragonisca fiebrigi* when exposed to chlorpyrifos, a commonly used insecticide. We employed a Hidden Markov Model (HMM) with MATLAB's Hidden Markov Model Toolkit (MATLABHTK) to analyze acoustic data from eight hives under both exposed and unexposed conditions, assessing the potential of acoustic monitoring as an indicator of pesticide-related stress. Initial analysis across multiple hives indicated moderate model performance. However, hive-specific analyses yielded higher performance in detecting pesticide exposure. Furthermore, the model accurately classified individual hives, suggesting the presence of a distinct acoustic 'fingerprint' for each hive. These findings advance the field of stingless bee bioacoustics and provide initial evidence that acoustic monitoring of stingless bee hives could be a useful and non-invasive tool to detect airborne pesticide contamination.

**Data availability statement:** All audio files are available from the zenodo database (DOI 10.5281/zenodo.15397732).

**Funding:** This study was financed in part by the Coordenação de Aperfeiçoamento de Pessoal de Nível Superior – Brasil (CAPES) – Finance Code 001 to AO. Funding for this project was also provided by NSF-IOS 2024026 to C.F.O. The funders had no role in study design, data collection and analysis, decision to publish, or preparation of the manuscript.

# Introduction

The ongoing global decline in both wild and managed bee populations is attributed to a combination of factors, including habitat loss, pathogens, climate change and the intensive use of pesticides [1,2]. While colony losses can be challenging to prevent, advancements in precision apiculture have led to the development of techniques for accurately monitoring hives and detecting stressors, such as *Varroa* mite infestations, queen absence, and, in some cases, exposure to pollutants [3–7]. These methods, which leverage machine learning models, are non-invasive and enable rapid, precise assessments of hive conditions, facilitating effective hive management and intervention.

One novel health metric of bee colonies is the colony's vibroacoustic signal. Indeed, bees produce various vibroacoustic signals through wing and body movements, and muscle contractions without wing movements at high frequency [8,9]. Bees use a range of different signals for communication [10–13]. Studies on honeybees have shown that the health, activity, and status of the hive can be determined by vibroacoustics [5,6,14–17]. In this species, sound patterns change in response to different stressors such as chemicals [6]. Honey bee vibrations have been processed with several machine learning models, including deep neural networks Internet of Things (IoT)-based solutions to detect swarming [18], and the use of Mel spectra, Mel-frequency cepstral coefficients (MFCCs), and Hilbert Huang Transform (HHT) to determine if a colony is queenright or queenless [17,19].

Hidden Markov models (HMM) are statistical models that are based on a Markov process in which the state of one event depends on the state of a previous one [20]. These models have been used in speech recognition, thermodynamics, and various pattern recognition applications [21]. The Hidden Markov Model Toolkit is a collection of source code that can be used within computing programs like MATLAB to generate Hidden Markov Models [22] for research focused on speech recognition. Although HMMs and HTK have been used to detect vertebrate sounds and movements [23–25] and HMMs have been explored as part of IoT solutions to detect swarming [18], the methodology has not otherwise been investigated for their ability to predict or assign various bee colony states.

Despite the rapid advancements in precision beekeeping and the demonstrated potential of these techniques to assess hive conditions, most studies have focused on *Apis mellifera* [14,26]. In contrast, stingless bees, a diverse group of eusocial insects native to Neotropical regions extending from Uruguay to central Mexico and also found in Africa, India, Southeast Asia, and Australia, remain underexplored in this field [27]. However, meliponiculture—the management of stingless bees—is gaining traction as both a source of income for families and a conservation strategy for these vital pollinators [28,29].

Stingless bees face significant threats, including pesticide exposure, to which they are more sensitive compared to honeybees [30]. For instance, larval gynes of *Plebeia droryana* exposed to the neonicotinoid insecticide imidacloprid exhibit abnormal post-eclosion behavior, leading to aggression and fatal attacks by worker bees [31]. Similarly, colonies of *Melipona quadrifasciata* exposed to neonicotinoid and pyrethroid pesticides show disruptions in social behavior and communication [32]. In *Tetragonisca angustula*, exposure to chlorpyrifos, cyflumetofen, and difenoconazole induces behavioral alterations among surviving bees [33].

Globally, the use of insecticides has risen over the last decades [34]. This trend is pronounced in Brazil, where agricultural activities expanded by 8.1% between 2000 and 2015, while insecticide use increased by an alarming 152% during the same period [35]. Although bees are highly susceptible to pesticide exposure and airborne contamination, the concentrations that harm bees often fall below thresholds for acute toxicity in humans [36,37].

Nonetheless, chronic, low-level exposure to these substances in humans can lead to irreversible health conditions and developmental issues [38,39]. Thus, monitoring bee populations offers a valuable bioindicator of environmental contamination [40,41].

In this study, we investigated the acoustic patterns of the stingless bee *Tetragonisca fiebrigi* using HMM as a means of detecting possible sound-based hive identification and pesticide-induced stress responses. Hives were exposed to chlorpyrifos, a widely used insecticide, to identify potential changes in their acoustic profiles. We recorded audio data during both undisturbed and exposure periods, aiming to detect shifts in acoustic signals indicative of stress, which could further support the potential use of stingless bee bioacoustics as bioindicators of airborne pesticide contamination.

## Materials and methods

### Colony details

The experiment was conducted in Mariana Pimentel, RS, Brazil (30°24'08.9"S, 51°35'06.5"W). Eight hives were included in the study. All hives met the following criteria: a) Presence of a queen; b) Recent brood cells, ensuring ongoing oviposition; c) Stocked honey and pollen reserves; d) No visible signs of disease or impairments.

Each hive was placed inside a plastic tunnel (three m long, one m wide, and 1.6 m high) with one side open to contain the pesticide and prevent it from spreading beyond the experimental area (Fig 1). The hives were three m apart from each other. The experiment consisted of two trials, both in 2024: the first took place from January 22 to 27 with four hives, and the second from March 25 to 31 with the remaining four hives. Each hive was recorded for two days undisturbed, then exposed to water spray on one day, pesticide spray on the next, and followed for two more days without disturbance. Subsequently, we describe each experimental step in detail.

### Vibroacoustic signal collection

Each hive was equipped with an AudioMoth® recorder, secured with a mosquito net and masking tape for protection (Fig 2). To avoid the known variability in bee activity during different times of the day, we selected a specific time window for all recordings. The devices were programmed to record from 12:30 PM to 5:30 PM, capturing 30-second audio clips every minute. The recordings were set at a 48 kHz sample rate with medium gain.

### Pesticide exposure

Initially, hives were recorded for two consecutive days without disturbance. On the third day, all hives were exposed to a water spray applied from one meter away from the hive entrance. On the fourth day, three hives received chlorpyrifos spray, while one control hive received only water spray. In the following days, the hives were once again recorded without disturbance (Fig 1).

To better reflect real-world conditions, we used a commercial chlorpyrifos formulation. Although the effects of adjuvants cannot be separated from those of the active ingredient in this approach, it enhances the relevance of our findings for biomonitoring applications. The exposure dose was based on the LC50 (0.0033 μg a.i./μL) for topical exposure in *Tetragonisca fiebrigi* [36]. The objective was not to kill the colonies, but to guarantee a scientifically supported effect on the bees that were topically exposed.

The spraying was performed using a pressure sprayer (Vonder®, 1.5 liters), pressurized by 13 pump strokes. The application was carried out one meter away from the entrance of

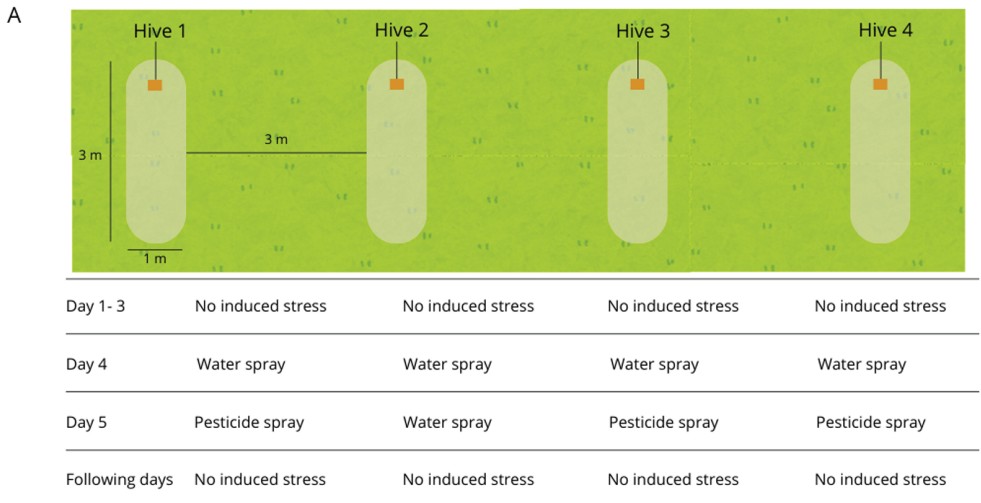

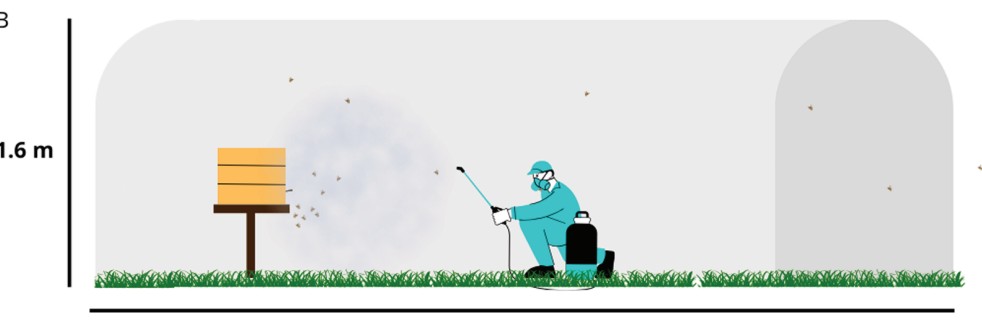

**Fig 1. Experimental design.** A) Tunnel placement and experimental layout. The tunnels were positioned three meters apart, with each containing a single hive. The diagram illustrates the activities conducted on each day of the experiment. B) Tunnel structure. Each tunnel measured three meters in length, one meter in width, and 1.6 meters in height. Spraying was performed in front of the hive using a pressure-based sprayer.

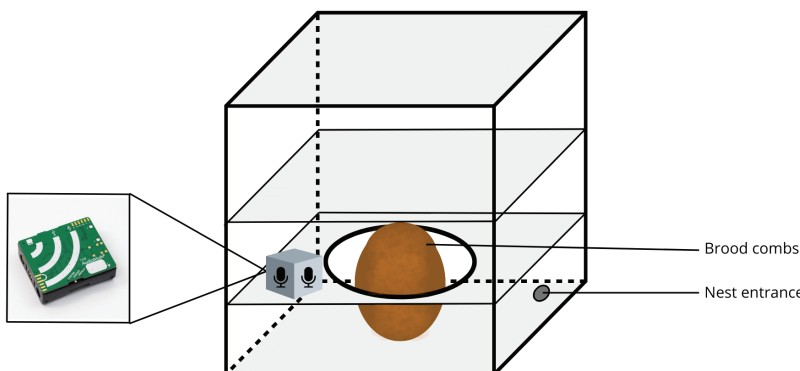

**Fig 2. Equipment installed inside the hive.** Schematic representation of the hive, showing the wooden modules, the brood combs located at the center, and the microphone positioned adjacent to the brood.

the hive and at a height of 60 centimeters. The liquids were sprayed in front of the hive for a duration of 18 seconds.

## File annotation

To create the training files, the .wav audio files from hives with known states were annotated by generating corresponding .txt files containing the states (as defined in Table 1). Initially, we annotated files to represent the chlorpyrifos exposure state, using data from all eight hives collectively (Table 1). Accordingly, audio files recorded during undisturbed conditions, including those involving water spray, were labeled as '0' (baseline). This included recordings from all hives, regardless of whether they were later exposed to chlorpyrifos or remained as controls, as long as no pesticide had been applied. The files corresponding to periods of 30 minutes to two and a half hours after pesticide exposure were labeled '1'. Subsequently, the same annotation process was applied, but each hive's dataset was analyzed individually in separate runs. Finally, we annotated files for basic colony identification across all studied hives. These label files were then utilized to train the Hidden Markov Model.

## Hidden Markov model development

We built a HMM according to Ranjard et al (2017) [25], with modifications. Briefly, an Oracle VirtualBox was installed on a Windows computer (32 GB RAM, 500 GB Hard Drive). Next, Ubuntu 20.04.2.0 LTS and MATLAB (version R2020b) were installed and the source code and samples files of a MATLAB Hidden Markov Model Toolkit (MATLABHTK) version 3.4.1 for Linux/Unix were downloaded (https://github.com/LouisRanjard/matlabHTK) and installed. Annotated sound files were used for training within the MATLABHTK framework and HMMs were produced for each state (2 HMMs for Chloropyrifos Exposure and 8 HMMs for Basic Hive Identification). Specifically, .wav files and corresponding label .txt files were analyzed using the train HTK command in MATLAB, which generated HMMs for each state. Next, .wav files without corresponding label files were analyzed using the recognise HTK command. Next, for each file, the state with the highest associated total time was determined and used for analysis of model accuracy (Fig 3).

## Performance metrics

Since the dataset was unbalanced, with more samples from no-exposure periods than from exposure periods, we randomly selected baseline samples to match the number of exposure samples. This resampling process was repeated 1,000 times, and the mean of the resulting metrics was calculated. This approach was applied to: (a) pesticide detection using the dataset from all hives combined, and (b) pesticide detection for each hive individually. For hive identification, both exposed and non-exposed periods were included, and the data was inherently balanced (equal number of samples per hive), so no resampling was necessary.

We evaluated the model's performance using several metrics to gain a detailed understanding of its effectiveness. Accuracy was analyzed to determine how often the model correctly

**Table 1. States used in file annotation.**

| State | State Levels |
|---|---|
| *ChlorpyrifosExposure* | 0 = baseline (including water spray); 1 = 30 min to 2.5 hrs after chlorpyrifos exposure |
| *BasicHiveIdentification* | 1-8, corresponding to each of the hives studied |

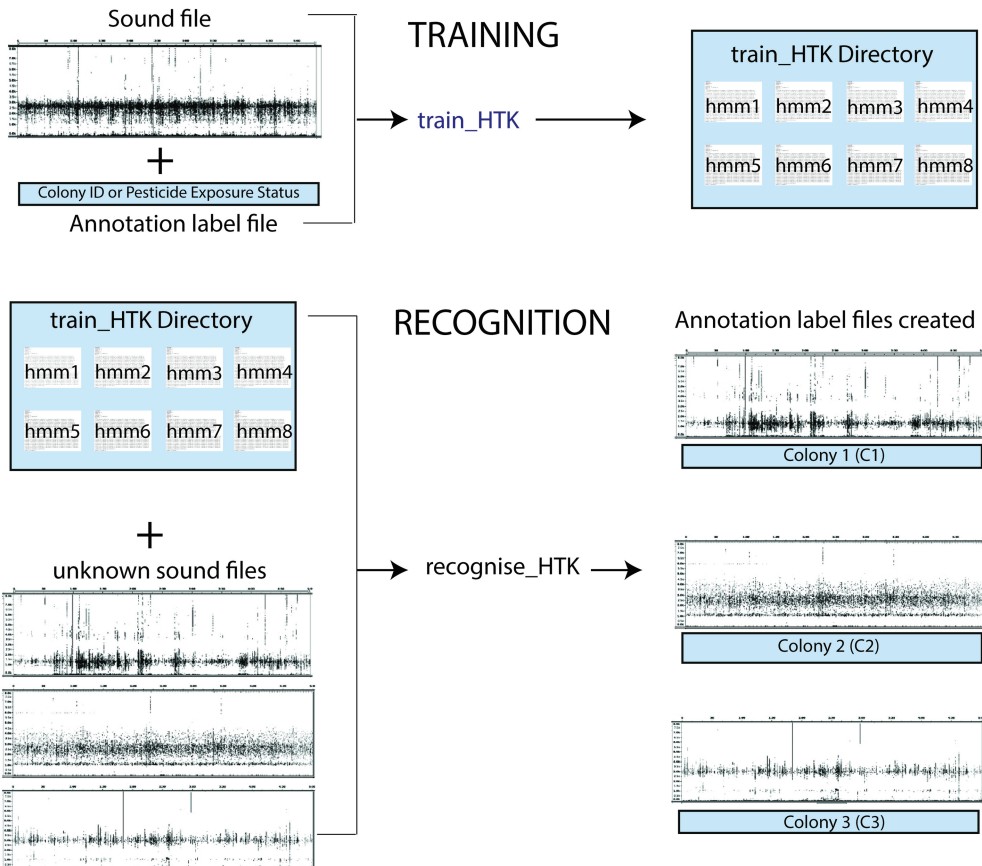

**Fig 3. Hidden Markov Model Training and Recognition Process with MATLABHTK.** Training with sound files and annotated label files was performed with the train HTK command while recognition was performed with the train HTK directory and unknown sound files with the recognise HTK command.

classified the data overall, providing a general measure of its reliability. Precision focused on the model's ability to accurately predict pesticide exposure periods from acoustic data, measuring the proportion of correct positive predictions (true positives) among all predictions of exposure. This helped assess how often the model avoided false alarms about pesticide exposure. Recall, also known as sensitivity, measured the model's ability to identify all actual periods of pesticide exposure. It quantified the proportion of true positives relative to the total number of actual exposure events, indicating how well the model captured all exposure periods without missing any.

Specificity, on the other hand, evaluated the model's ability to correctly identify non-exposure periods. It calculated the proportion of true negatives (correctly identified non-exposure periods) among all actual non-exposure events, helping assess how effectively the model avoided incorrectly labeling non-exposure periods as exposure.

Finally, the F1 score provided a balanced evaluation by combining precision and recall into a single metric. Calculated as the harmonic mean of these two measures, the F1 score offered a comprehensive view of the model's ability to detect true exposure periods while minimizing false positives.

## Results and discussion

### Initial observations

During the application of both water and the pesticide, guard bees and foragers at the nest entrance came into direct contact with the sprayed liquid. Following each exposure, foraging activity was temporarily interrupted, and all guard bees retreated into the hive. In the case of pesticide application, this behavior may have contributed to the substance's entry into the colony. For both treatments, normal activity resumed approximately six minutes after the disturbance. Liquid droplets were deposited on the wooden hive surface, with a few also reaching the ground. No dead bees were observed at the nest entrance or on the surrounding area, and the droplets evaporated shortly after application.

### Pesticide exposure detection

Our results demonstrate that HMM can effectively classify when *Tetragonisca fiebrigi* hives are exposed to the chlorpyrifos insecticide at the CL50 for topical exposure. However, the model's performance was lower when using the dataset from all eight hives combined compared to when hive-specific datasets were used (Table 2).

Notably, when analyzing the combined dataset, the specificity—reflecting the model's ability to correctly classify non-exposure states—was higher than recall, which measures the correct classification of exposure states. The model showed a high precision for exposure classification, indicating that most files classified as exposure were correctly labeled. However, it misclassified a significant portion of exposure files as non-exposure, leading to a recall value slightly above random chance. This highlights a limitation in detecting exposure events under the current dataset conditions.

It is known that variability in hive conditions, such as disease presence, environmental stressors, and resource availability, must be considered when assessing bee toxicological responses [42]. The hives in this study maintained standardized health conditions and were recorded under similar environmental parameters. Therefore, the results may have been influenced by two nonexclusive factors: (i) variability in the acoustic response to pesticide exposure between colonies, and/or (ii) intrinsic acoustic differences between colonies, regardless of exposure status, reflecting a kind of colony-specific acoustic identity. To investigate this, the model was subsequently trained and tested using data from each hive individually.

When the model was trained using data from individual hives—analyzing each hive separately—performance varied between hives (Table 2) but consistently exceeded the performance achieved when using the combined dataset of all hives. Notably, the F1 score, which reflects the balance between precision and recall, was higher than 0.83 for all individual hive exposure classifications, compared to 0.61 for the combined dataset. This improvement was

**Table 2. HMM Metrics in Pesticide Exposure Detection Task.**

| Hive | Accuracy | 95% CI | Specificity | Precision | Recall | F1 Score |
|------|----------|--------|-------------|-----------|--------|----------|
| All | 0.6757 | (0.6468, 0.7036) | 0.8420 | 0.7632 | 0.5093 | 0.6109 |
| 1 | 0.8681 | (0.8102, 0.9136) | 0.8352 | 0.8454 | 0.9011 | 0.8723 |
| 3 | 0.8297 | (0.7670, 0.8812) | 0.7634 | 0.7843 | 0.8989 | 0.8377 |
| 4 | 0.8626 | (0.8039, 0.9091) | 0.8901 | 0.8837 | 0.8352 | 0.8588 |
| 5 | 0.8516 | (0.7915, 0.8999) | 0.7473 | 0.7909 | 0.9560 | 0.8657 |
| 7 | 0.8352 | (0.7731, 0.8859) | 0.7143 | 0.7699 | 0.9560 | 0.8529 |
| 8 | 0.8077 | (0.7428, 0.8622) | 0.6374 | 0.7295 | 0.9780 | 0.8357 |

largely driven by higher recall values, indicating more accurate detection of exposure states. These findings suggest that each hive may exhibit a distinct acoustic pattern, a hypothesis further tested in subsequent analyses.

The fact that several exposure events were not identified by the classification model trained with data from all hives could indicate that the bees' response to chlorpyrifos commercial formula exposure may not have been sufficiently intense to trigger detectable acoustic alterations. In a few isolated instances, the exposure could have induced more pronounced behavioral changes, resulting in detectable and correctly classified acoustic shifts. However, this interpretation is challenged by the high performance metrics obtained when the model was trained individually for each hive.

In addition, these findings highlight that the use of baseline recordings of the same hive prior to stress provides a more reliable control than data from a separate control hive recorded at the same time. This approach enhances sensitivity to subtle stress-related acoustic changes that might otherwise be masked by inter-colony variability. However, the extent to which the number of hives included in the analysis influences this outcome remains unclear. This aspect warrants further investigation, although several acoustic studies in bees have been conducted using a small number of colonies (between one and six beehives) [26].

We acknowledge that the requirement for a personalized model trained on data from a specific hive may represent a limitation of the proposed approach. While improved performance was observed with individualized datasets, such models can be costly and may not scale effectively. Similar findings have been reported in other domains, for instance, in human emotion recognition, where personalized models showed better performance than general models [43,44]. Whether the increased performance of personalized models justifies their higher implementation cost compared to the broader applicability of generalized models remains an open question in computer science.

Previous researchers have successfully analyzed hive responses to chemicals through sound in honeybees. By recording the sound of four hives, oral exposure of bees to syrup laced with acetone or ethyl acetate was classified using three machine learning algorithms: k-nearest neighbors (KNN), random forest (RF), and support vector machine (SVM) [45]. All algorithms achieved over 90% accuracy. Using the dataset of three honeybee hives exposed to common air pollutants such as acetone, trichloromethane, glutaric dialdehyde, and ethyl ether introduced inside the nests, SVM outperformed the others with an average classification accuracy of 93.7%, compared to 83.8% for KNN and 83.6% for RF in analyzing audio samples [46]. Additionally, flight activity data (rather than sound) was successfully used to detect exposure to a mixture of pesticides [47]. Beehive acoustic responses to trichloromethane-laced air were also assessed using soundscape indices [6].

Our results represent a first step toward using stingless bees as bioindicators of airborne pesticide contamination through a non-invasive method that eliminates the need for laboratory analysis. This approach is essential not only for protecting pollinators and biodiversity, but also to reduce the risk of chronic human exposure to low doses of pesticides, which often go unnoticed initially but can lead to serious long-term health effects [38,39].

## Hive identification

Using the entire dataset of each hive (including both undisturbed and pesticide exposure moments, except for the control hives), the HMMs successfully classified the hives from which the sounds were collected (Table 3).

Despite the hives sharing similar conditions, such as abundant honey and pollen resources and high foraging activity, distinct acoustic variations were observed. While factors like box

**Table 3. Model accuracy in classifying audio files from each hive.**

| Hive 1 | Hive 2 | Hive 3 | Hive 4 | Hive 5 | Hive 6 | Hive 7 | Hive 8 |
|--------|--------|--------|--------|--------|--------|--------|--------|
| 0.97321 | 0.93898 | 0.96529 | 0.99071 | 0.9503 | 0.9509 | 0.8961 | 0.9820 |

material, box size, and nest structure could influence hive acoustics, these variables were standardized across the experiment. Variability among the audio recorders is also unlikely to explain the differences, as the same four devices were used consistently in both trials. Our results further revealed that hives originating from the same location (1 and 5; 4 and 8; 2, 3, 5, and 6) did not exhibit classification errors (Fig 4), indicating distinct acoustic profiles for each hive. Moreover, even when four hives were recorded simultaneously under identical environmental conditions, the model maintained high classification accuracy, effectively distinguishing between the nests. These findings strongly suggest that each hive of this stingless bee species exhibits a unique acoustic signature.

Although bioacoustic research in precision apiculture has not primarily focused on this idea, a similar concept emerged in 2021, when specific sound patterns were identified in *Apis mellifera* hives [48]. In that study, specific buzz patterns were used to characterize individual colonies, leading to the proposal of an "acoustic fingerprint" method for colony identification. In a related effort, researchers employing multi-label classification models were able to identify individual honeybee hives based on their sound profiles with 90% accuracy [49]. The authors suggested that future work should focus on developing hive-specific models to monitor colony-level events and stressors.

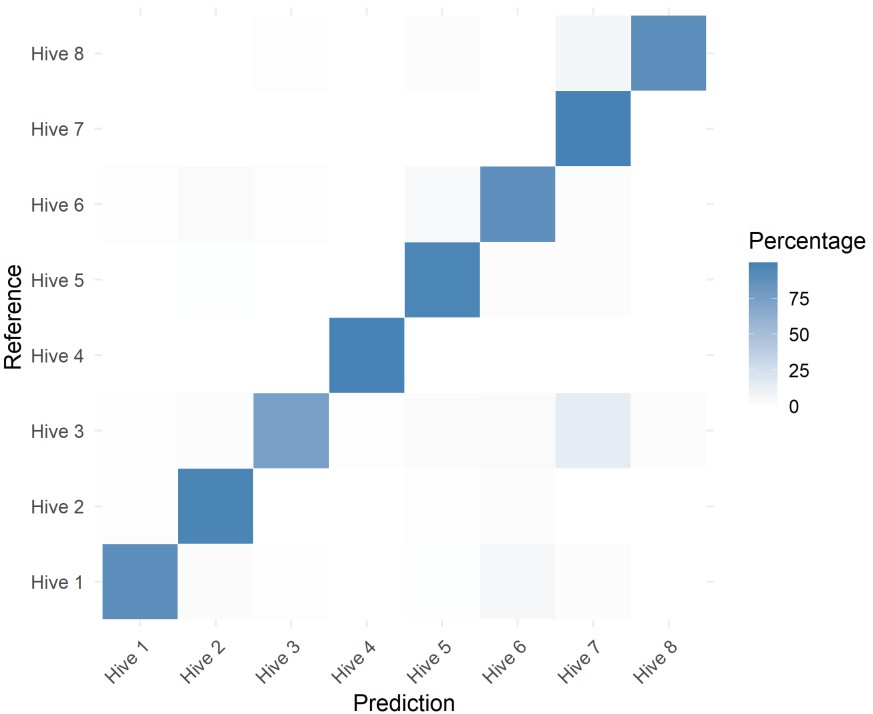

**Fig 4. Confusion Matrix Heatmap.** The x-axis represents the model's predictions, while the y-axis represents the actual classifications. The color intensity increases as the percentage of correctly classified data approaches 100%.

## Conclusion

This study successfully demonstrated the potential of Hidden Markov Models (HMM) in detecting exposure of *Tetragonisca fiebrigi* hives to the pesticide chlorpyrifos, highlighting the value of bioacoustic approaches in monitoring stressors in stingless bees. Our findings revealed that hive-specific datasets significantly enhance model performance, suggesting that individualized acoustic patterns could be a critical factor in precision meliponiculture. This study serves as an important step forward in the bioacoustics of stingless bees and provides the first evidence supporting the use of acoustics as an indicator of pesticide exposure in this taxa. While our results are robust and consistent with sample sizes commonly used in bee bioacoustics research, increasing the number of hives and testing additional pesticides would provide a more comprehensive understanding of the method's applicability. Future research should also investigate the scalability of this approach to other stingless bee species and pollinators. By bridging the fields of bioacoustics, machine learning, and conservation, this work lays the groundwork for innovative, non-invasive monitoring tools to support pollinator, biodiversity and human health.

## Acknowledgments

AO thanks Ilvonaldo Lopes Otesbelgue and Rubim Benoni Stein for their contributions in constructing the structure used for conducting the experiments. We thank Dustin Frank for his technical assistance with the vibroacoustics interpretation.

## Author contributions

**Conceptualization:** Alex Otesbelgue, Amara Jean Orth, Carol Anne Fassbinder-Orth, Betina Blochtein, Maria João Ramos Pereira.

**Data curation:** Alex Otesbelgue, Carol Anne Fassbinder-Orth.

**Formal analysis:** Alex Otesbelgue, Carol Anne Fassbinder-Orth.

**Funding acquisition:** Alex Otesbelgue, Carol Anne Fassbinder-Orth.

**Investigation:** Alex Otesbelgue, Amara Jean Orth, Chandler David Fong, Carol Anne Fassbinder-Orth.

**Methodology:** Alex Otesbelgue, Carol Anne Fassbinder-Orth.

**Project administration:** Alex Otesbelgue, Carol Anne Fassbinder-Orth, Betina Blochtein, Maria João Ramos Pereira.

**Resources:** Alex Otesbelgue, Carol Anne Fassbinder-Orth.

**Software:** Alex Otesbelgue, Chandler David Fong, Carol Anne Fassbinder-Orth.

**Supervision:** Alex Otesbelgue, Betina Blochtein, Maria João Ramos Pereira.

**Validation:** Alex Otesbelgue, Amara Jean Orth, Chandler David Fong, Carol Anne Fassbinder-Orth, Betina Blochtein, Maria João Ramos Pereira.

**Visualization:** Alex Otesbelgue, Carol Anne Fassbinder-Orth.

**Writing – original draft:** Alex Otesbelgue, Carol Anne Fassbinder-Orth.

**Writing – review & editing:** Alex Otesbelgue, Amara Jean Orth, Chandler David Fong, Carol Anne Fassbinder-Orth, Betina Blochtein, Maria João Ramos Pereira.

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
