## [Decision Letter · Decision Letter 0]

24 Mar 2025

PONE-D-25-08785Hidden Markov Model for Acoustic Pesticide Exposure Detection and Hive Identification in Stingless BeesPLOS ONE

Dear Dr. Otesbelgue,

Thank you for submitting your manuscript to PLOS ONE. After careful consideration, we feel that it has merit but does not fully meet PLOS ONE’s publication criteria as it currently stands. Therefore, we invite you to submit a revised version of the manuscript that addresses the points raised during the review process.

Introduction needs improvement with pointwise suggestions by the reviewers.

Materials and methods should be clear and answer the questions raised by the reviewer(s) especially the small sample size and number of colonies.

Results with critical variation in bees’ response of different colonies need specific address for authentic use in further studies or their utilization. There is also a limited critical discussion which need to be improved.

We look forward to receiving your revised manuscript.

Kind regards,

Munir Ahmad, PhD

Academic Editor

PLOS ONE

Journal Requirements:

2**.** In your Methods section, please provide additional information regarding the permits you obtained for the work. Please ensure you have included the full name of the authority that approved the field site access and, if no permits were required, a brief statement explaining why.

“This study was financed in part by the Coordenação de Aperfeiçoamento de Pessoal de Nível Superior – Brasil (CAPES) – Finance Code 001” to AO. Funding for this project was also provided by NSF-IOS 2024026 to C. Fassbinder-Orth.”

**Additional Editor Comments:**

Introduction needs improvement with pointwise suggestions by the reviewers.

Materials and methods should be clear and answer the questions raised by the reviewer(s) especially the small sample size and number of colonies.

Results with critical variation in bees’ response of different colonies need specific address for authentic use in further studies or their utilization. There is also a limited critical discussion which need to be improved.

Reviewers' comments:

Reviewer's Responses to Questions

**Comments to the Author**

1. Is the manuscript technically sound, and do the data support the conclusions?

Reviewer #1: Yes

Reviewer #2: Yes

2. Has the statistical analysis been performed appropriately and rigorously? 

Reviewer #1: I Don't Know

Reviewer #2: Yes

3. Have the authors made all data underlying the findings in their manuscript fully available?

Reviewer #1: No

Reviewer #2: Yes

4. Is the manuscript presented in an intelligible fashion and written in standard English?

Reviewer #1: Yes

Reviewer #2: Yes

5. Review Comments to the Author

Reviewer #1: Otesbelgue et al. present a manuscript on the use of acoustics as a toxicovigilance tool for pesticide exposure detection on stingless bees. The results of the study are very promising and of interest to the readers of PLOSONE. In general, the text is well written and almost devoid of typos. However, the text lacks clarity in some important methodological aspects and the study is very limited in the number of hives evaluated (6 experimental hives and 2 control hives). Also, the figures are of very low resolution. The discussion is very limited. And should be expanded. Therefore, I recommend a major revision.

Here are my comments and specific questions:

Line 73. The Figure is a bit deceptive as the length of the tunnel seems longer than 3 m.

Line 73 Figure 1. The resolution of the image should be improved. This applies to all figures.

Line 76 Did you use the pure molecule? Or a commercial pesticide? If you used the pure molecule, how did you dissolve it in water? If you used a commercial pesticide, how can we be sure the effect is only from the chlorpyrifos and not its adjuvants?

Line 84. I think the design is flawed. The study only used 1 control hive per replica. The problem with having only one control hive is that if for some reason, not related to the experiment, that hive behaved differently you would immediately assume it was due to treatment. I think this is a major drawback of the study. A better design would have been 2 control vs 2 treated hives.

Line 103. Did you remove from the recordings the time lapse of pesticide or water application. This fiscal disturbance could bias your results. How can we be sure the bees are not affected by the smell of the adjuvants rather than the toxicity of the active ingredient. I suggest removing from the recordings a certain time lapse when the application was performed to avoid this possible source of noise.

Table 2. Here you use C1-C8 and in another figure you use H1-H8.

Results. Are there any acoustic metrics that differed amongst treatments?

Reviewer #2: I have carefully reviewed the manuscript titled ‘Hidden Markov Model for Acoustic Pesticide Exposure Detection and Hive Identification in Stingless Bees’ and would like to provide my general feedback regarding its content and suitability for publication in PLOS One. The manuscript investigates the use of bioindicators for airborne pesticide contamination (chlorpyrifos) in Tetragonisca fiebrigi hives. This is an innovative approach for toxicological studies involving stingless bees and holds promise as a non-invasive strategy for monitoring hive conditions. The topic of the manuscript has great potential. Below are a few suggestions that could help the authors enhance the clarity and readability of the paper.

Introduction

The introduction is well-written. I followed the flow of ideas and the explanations regarding the topic. The text is clear, easy to read, and understand.

Line 49: There seems to be an extra bracket, likely a typo.

Materials and methods

The method is well presented and facilitates reproducibility. Below, I will address some general concerns.

Figure 1: The image quality in the PDF file is very poor. This could be an issue related to the submission process. However, the content of Figure 1A is not readable. There are symbols such as droplets and triangles, but their meaning is unclear, making it difficult to evaluate this part.

Lines 74 and 75: The method applied appears accurate. However, this sentence raised some questions, such as whether the sound was recorded from the same colony both before and after insecticide exposure. These questions are addressed later in the text, but I suggest briefly noting that this topic will be discussed in more detail later.

Line 79: The sound was recorded between 12:30 PM and 5:30 PM. Why were this time frame chosen? Is this stingless bee species more active during this period?

Line 90: Were bees flying outside the hive entrance during the insecticide spray application? Were only the bees protecting the hive exposed to the insecticide particles? More importantly, could the particles have settled on the wooden hive boxes or fallen to the ground? The main question is whether the number of exposed bees was sufficient to cause any internal alterations.

Figure 3: Providing more detailed explanations about Figure 3 could improve the reader's understanding of the meaning of each set.

Results and Discussion

Line 184: The hypothesis of a unique "fingerprint" for each colony is interesting. However, a question arises from this statement: if each colony has a unique sound, would evaluating exposure through bioindicators of airborne pesticide contamination become more challenging? While the authors highlight the advantages of applying this method in lines 174-179, I wonder: what about the disadvantages? The unique sound described for each colony seems to present a challenge for the proposed analysis method.

6. PLOS authors have the option to publish the peer review history of their article (what does this mean?). If published, this will include your full peer review and any attached files.

Reviewer #1: No

Reviewer #2: No

---

## [Author Response · Author response to Decision Letter 1]

15 May 2025

Dear Reviewer 1,

We sincerely thank you for your thoughtful contributions, which have significantly improved the quality of our manuscript, especially by encouraging us to provide more detailed methodological descriptions to ensure that readers can fully understand our procedures.

We have improved the image resolution to meet the required standards. Additionally, we addressed important points regarding the number of hives used in the study, discussed our perspectives, and compared our sample size with those of other studies involving contaminant exposure and honeybee bioacoustics. We also emphasized in the text that increasing the number of hives in future studies could bring new insights and help answer some outstanding questions.

Furthermore, we incorporated additional ideas and references into the discussion section, which not only made the manuscript more comprehensive but also allowed us to highlight key aspects of our results and interpret them within the broader context of existing scientific knowledge.

We truly appreciate your feedback, which helped us strengthen our work.

Here are my comments and specific questions:

Line 73. The Figure is a bit deceptive as the length of the tunnel seems longer than 3 m.

We revised the figures to ensure consistent scaling, maintaining the same length for the tunnels and accurately representing the three-meter distance between them. Additionally, we included the tunnel length and width measurements to provide clearer spatial information.

Line 73 Figure 1. The resolution of the image should be improved. This applies to all figures.

We apologize for the low image resolution observed after uploading to the submission system. We have since reviewed and corrected the issue, ensuring that the figures now meet the required resolution standards.

Line 76 Did you use the pure molecule? Or a commercial pesticide? If you used the pure molecule, how did you dissolve it in water? If you used a commercial pesticide, how can we be sure the effect is only from the chlorpyrifos and not its adjuvants?

We used the commercial formulation of chlorpyrifos, the same product referenced in Dorneles et al. (2017), which we used to define the dose corresponding to the LC₅₀ for topical exposure. The formulation contains 48% of the active ingredient, and this proportion was carefully considered in the preparation of the solution used for exposure.

Our choice to use the commercial product rather than the pure molecule was deliberate and grounded in ecological relevance. In real-world scenarios, bees are exposed not to isolated active ingredients, but to complete pesticide formulations, including solvents, surfactants, and other adjuvants. We acknowledge that using the commercial formulation means the observed effects could result from chlorpyrifos, its adjuvants, or a combination of both. However, for our study, this does not represent a limitation, but rather a necessary element of ecological validity. One of our key research questions was to assess whether acoustic changes in stingless bee colonies can serve as indicators of airborne pesticide exposure. To test this hypothesis in a realistic manner, it was essential to use the product as it is applied in the field. By exposing bees to the actual formulation used in agriculture, we increase the relevance and potential applicability of our findings for biomonitoring programs.

We appreciate the reviewer’s comment and understand the concern. To clarify our choice for the readers, we added the following statement in the "Pesticide Exposure" subsection of the Materials and Methods section:

" To better reflect real-world conditions, we used a commercial chlorpyrifos formulation. Although the effects of adjuvants cannot be separated from those of the active ingredient in this approach, it enhances the relevance of our findings for biomonitoring applications."

Additionally, we explained the rationale behind the concentration used:

"Our objective was not to cause colony mortality, but rather to ensure a scientifically supported sublethal effect on the bees that were topically exposed."

Line 84. I think the design is flawed. The study only used 1 control hive per replica. The problem with having only one control hive is that if for some reason, not related to the experiment, that hive behaved differently you would immediately assume it was due to treatment. I think this is a major drawback of the study. A better design would have been 2 control vs 2 treated hives.

We appreciate the reviewer’s concern regarding the use of a single control hive per replicate. We agree that, in many experimental designs, using multiple control units per treatment group is a way to control natural variability and increase robustness. However, our study was designed with a different strategy in mind, guided by the specific goals of the research.

All hives, whether exposed or not, experienced the same environmental conditions, and we recorded each one over multiple days, capturing a range of natural behavioral and acoustic variations, such as increased foraging activity or exposure to non-toxic stressors like water spray. All these events were included in the "non-exposure" category to help train the model to distinguish between pesticide-related and unrelated changes. This means that our control group (labeled as 0) in the first run of the algorithm included data from eight different hives.

We acknowledge that this point may not have been sufficiently clear in the original text. Therefore, we added a clarification in the "File Annotation" subsection of the Materials and Methods section.

“Accordingly, audio files recorded during undisturbed conditions, including those involving water spray, were labeled as '0' (baseline). This included recordings from all hives, regardless of whether they were later exposed to chlorpyrifos or remained as controls, as long as no pesticide had been applied. The files corresponding to periods of 30 minutes to two and a half hours after pesticide exposure were labeled '1'.”

Although we used recordings from different days and from different hives, we observed that the model's performance when trained on the overall dataset was lower than when it was trained separately on each hive's dataset. We suggest that this result may be due to natural acoustic variability among hives.

To help the reader follow this idea, we added the following explanation to the Results and Discussion section:

“Despite the hives exhibiting standardized health conditions and being recorded under the same environmental parameters, the results may have been influenced by two non-exclusive factors: (i) variability in the acoustic response to pesticide exposure between colonies, and/or (ii) intrinsic acoustic differences between colonies, regardless of exposure status, reflecting a kind of colony-specific acoustic identity. To investigate this, the model was subsequently trained and tested using data from each hive individually.”

And after presenting the classification results using models trained individually for each hive, we added the following explanation to the text:

“In addition, these findings highlight that the use of baseline recordings of the same hive prior to stress provides a more reliable control than data from a separate control hive recorded at the same time. This approach enhances sensitivity to subtle stress-related acoustic changes that might otherwise be masked by inter-colony variability.”

Line 103. Did you remove from the recordings the time lapse of pesticide or water application. This fiscal disturbance could bias your results. How can we be sure the bees are not affected by the smell of the adjuvants rather than the toxicity of the active ingredient. I suggest removing from the recordings a certain time lapse when the application was performed to avoid this possible source of noise.

We chose not to exclude the time window during or immediately after the application of either the pesticide or water. To control potential disturbances caused by the physical presence of the applicator, the act of spraying, or the noise produced during spraying, we applied water in front of all hives one day prior to pesticide exposure. This approach allowed us to mimic the disturbance event without introducing the chemical factor. Recordings associated with water spray were labeled as '0' (baseline), ensuring that the model would not be biased by the disturbance itself.

Moreover, exploratory analyses from a pilot experiment (not included in the dataset presented in this manuscript to preserve the experimental design) suggested that detectable colony-level responses to chlorpyrifos exposure emerged approximately 30 minutes after application. That is why we performed the file annotation as described in Table 1.

As mentioned before, to address the inability to distinguish between the effects, we added phrases in the pesticide exposure subsection of the Materials and Methods section that clarify this point.

Table 2. Here you use C1-C8 and in another figure you use H1-H8.

We have now removed the 'C' designation, retaining only the hive number. Additionally, all figures and tables have been updated to consistently refer to 'hive' rather than 'colony' to improve clarity and understanding.

Results. Are there any acoustic metrics that differed amongst treatments?

We did not apply feature extraction methods; therefore, our methodology does not support assumptions regarding the specific acoustic characteristics underlying the model's predictions. Future studies incorporating feature extraction or feature engineering could address this limitation and identify acoustic metrics that distinguish between treatments or individual hives.

Reviewer #2: I have carefully reviewed the manuscript titled ‘Hidden Markov Model for Acoustic Pesticide Exposure Detection and Hive Identification in Stingless Bees’ and would like to provide my general feedback regarding its content and suitability for publication in PLOS One. The manuscript investigates the use of bioindicators for airborne pesticide contamination (chlorpyrifos) in Tetragonisca fiebrigi hives. This is an innovative approach for toxicological studies involving stingless bees and holds promise as a non-invasive strategy for monitoring hive conditions. The topic of the manuscript has great potential. Below are a few suggestions that could help the authors enhance the clarity and readability of the paper.

We thank Reviewer #2 for their constructive comments. We appreciate the recognition of the manuscript's potential and have carefully considered all suggestions to improve the clarity, readability, and scientific rigor of the work. In response, we corrected minor typographical issues, clarified aspects of the experimental design and methodology, improved figure quality, and expanded the discussion to better address the behavioral observations during exposure and the implications of colony-specific acoustic patterns. We believe these revisions have significantly strengthened the manuscript and addressed the reviewer’s concerns.

Introduction

The introduction is well-written. I followed the flow of ideas and the explanations regarding the topic. The text is clear, easy to read, and understand.

Line 49: There seems to be an extra bracket, likely a typo.

We have removed the extra bracket on line 49 and corrected the typo error.

Materials and methods

The method is well presented and facilitates reproducibility. Below, I will address some general concerns.

Figure 1: The image quality in the PDF file is very poor. This could be an issue related to the submission process. However, the content of Figure 1A is not readable. There are symbols such as droplets and triangles, but their meaning is unclear, making it difficult to evaluate this part.

We improved the image quality. In Figure 1, we ensured that the scale was accurately represented in the drawings and removed additional symbols that could hinder the clear interpretation of the image.

Lines 74 and 75: The method applied appears accurate. However, this sentence raised some questions, such as whether the sound was recorded from the same colony both before and after insecticide exposure. These questions are addressed later in the text, but I suggest briefly noting that this topic will be discussed in more detail later.

In this section, we added a brief description of the experimental design: “Each hive was recorded for two days undisturbed, then exposed to water spray on one day, pesticide spray on the next, and followed for two more days without disturbance. Subsequently, we describe each experimental step in detail.”

Line 79: The sound was recorded between 12:30 PM and 5:30 PM. Why were this time frame chosen? Is this stingless bee species more active during this period?

Stingless bees are known to be more active during the morning hours. However, they continue to exhibit intense out-of-nest activities throughout daylight hours, provided that the temperature remains favorable. We selected the exposure time based on practical considerations, while ensuring that the bees were still actively foraging.

We added in line 79-80: “To avoid the known variability in bee activity during different times of the day, we selected a specific time window for all recordings.”

Line 90: Were bees flying outside the hive entrance during the insecticide spray application? Were only the bees protecting the hive exposed to the insecticide particles? More importantly, could the particles have settled on the wooden hive boxes or fallen to the ground? The main question is whether the number of exposed bees was sufficient to cause any internal alterations.

In the Methods section, line 92, we revised the sentence to:

“The objective was not to kill the hives, but to ensure an effect on the topically exposed bees.”

In the Results and Discussion sections, line 149-158, we added a subsection to describe the behaviors observed during and shortly after exposure.

“Initial Observations

During the application of both water and the pesticide, guard bees and foragers at the nest entrance came into direct contact with the sprayed liquid. Following each exposure, foraging activity was temporarily interrupted, and all guard bees retreated into the hive. In the case of pesticide application, this behavior may have contributed to the substance's entry into the colony. For both treatments, normal activity resumed approximately six minutes after the disturbance. Liquid droplets were deposited on the wooden hive surface, with a few also reaching the ground. No dead bees were observed at the nest entrance or on the surrounding area, and the droplets evaporated shortly after application.”

In results and discussion lines 188-194, we raised this concern and explored this idea:

“The fact that several exposure events were not identified by the classification model trained with data from all hives could indicate that the bees' response to chlorpyrifos commercial formula exposure may not have been sufficiently intense to trigger detectable acoustic alterations. In a few isolated instances, the exposure could have induced more pronounced behavioral changes, resulting in detectable and correctly classified acoustic shifts. However, this interpretation is challenged by the high performance metrics obtained when the model was trained individually for each hive.”

Figure 3: Providing more detailed explanations about Figure 3 could improve the reader's understanding of the meaning of each set.

Line 120: Annotated sound files were used for training within the MATLABHTK framework and HMMs were produced for each state (2 HMMs for Chloropyrifos Exposure and 8 HMMs for Basic Hive Identification). Specifically, .wav files and corresponding label .txt files were analyzed using the train_HTK command in MATLAB, which generated HMMs for each state. Next, .wav files without corresponding label files were analyzed using the recognise_HTK command. Next, for each file, the state with the highest associated total time was determined and used for analysis of model accuracy (Fig 3).

Results and Discussion

Line 184: The hypothesis of a unique "fingerprint" for each colony is interesting. However, a question arises fro

---

## [Editor Report · Decision Letter 1]

18 May 2025

Hidden Markov Model for Acoustic Pesticide Exposure Detection and Hive Identification in Stingless Bees

PONE-D-25-08785R1

Dear Dr. Otesbelgue,

We’re pleased to inform you that your manuscript has been judged scientifically suitable for publication and will be formally accepted for publication once it meets all outstanding technical requirements.

Kind regards,

Munir Ahmad, PhD

Academic Editor

PLOS ONE
---

## [Editor Report · Acceptance letter]

PONE-D-25-08785R1

PLOS ONE

Dear Dr. Otesbelgue,

I'm pleased to inform you that your manuscript has been deemed suitable for publication in PLOS ONE. Congratulations! Your manuscript is now being handed over to our production team.

Kind regards,

on behalf of

Dr. Munir Ahmad

Academic Editor

PLOS ONE